# Machine-Learning Techniques for Feature Selection and Prediction of Mortality in Elderly CABG Patients

**DOI:** 10.3390/healthcare9050547

**Published:** 2021-05-07

**Authors:** Yen-Chun Huang, Shao-Jung Li, Mingchih Chen, Tian-Shyug Lee, Yu-Ning Chien

**Affiliations:** 1Graduate Institute of Business Administration, College of Management, Fu Jen Catholic University, New Taipei City 24205, Taiwan; hivicky92@gmail.com; 2Artificial Intelligence Development Center, Fu Jen Catholic University, New Taipei City 242062, Taiwan; 151294@mail.fju.edu.tw; 3Cardiovascular Research Center, Wan Fang Hospital, Taipei Medical University, Taipei 242, Taiwan; leeshaojung@gmail.com; 4Taipei Heart Institute, Taipei Medical University, Taipei 242, Taiwan; 5Department of Surgery, School of Medicine, College of Medicine, Taipei Medical University, Taipei 242, Taiwan; 6Division of Cardiovascular Surgery, Department of Surgery, Wan Fang Hospital, Taipei Medical University, Taipei 242, Taiwan; 7Master Program of Big Data Analysis in Biomedicine, College of Medicine, Fu Jen Catholic University, New Taipei City 242062, Taiwan

**Keywords:** National Health Insurance Research Database, NHIRD, older adults, CABG, machine learning, overall survival prediction, feature selection

## Abstract

Coronary artery bypass surgery grafting (CABG) is a commonly efficient treatment for coronary artery disease patients. Even if we know the underlying disease, and advancing age is related to survival, there is no research using the one year before surgery and operation-associated factors as predicting elements. This research used different machine-learning methods to select the features and predict older adults’ survival (more than 65 years old). This nationwide population-based cohort study used the National Health Insurance Research Database (NHIRD), the largest and most complete dataset in Taiwan. We extracted the data of older patients who had received their first CABG surgery criteria between January 2008 and December 2009 (*n* = 3728), and we used five different machine-learning methods to select the features and predict survival rates. The results show that, without variable selection, XGBoost had the best predictive ability. Upon selecting XGBoost and adding the CHA2DS score, acute pancreatitis, and acute kidney failure for further predictive analysis, MARS had the best prediction performance, and it only needed 10 variables. This study’s advantages are that it is innovative and useful for clinical decision making, and machine learning could achieve better prediction with fewer variables. If we could predict patients’ survival risk before a CABG operation, early prevention and disease management would be possible.

## 1. Introduction

Advancing age leads to markedly increasing coronary artery disease (CAD), a common heart disease and the leading global cause of mortality [1], significantly increasing the global healthcare burden [2]. Coronary artery bypass grafting (CABG) is an efficient treatment for patients with CAD in myocardial revascularization [3]. The risk of CABG surgery is approximately 1–3%. CABG is also high-cost surgery [4]. In recent years, various studies evaluated CABG risk on survival rate, medical cost, and follow-up of different CAD treatment strategies [3,4,5,6,7,8].

However, there is no complete research using an extensive database to build an integral machine-learning model for predicting and evaluating which risk factors could preoperatively affect older adults’ survival rate. Thus, this research used the National Health Insurance Research Database (NHIRD), with a sufficiently large data sample of Taiwan, which provided all real and large healthcare data, including patients’ original clinical records, treatments, inhospital expenditures, and diagnosis codes. In addition to the patients’ basic characteristics and disease history, we used variables before one year and during the operation as predictive indicators. Therefore, if we could predict patients’ mortality risk before a CABG operation, take early prevention and disease management for those high-risk patients would be possible. Our studies used multistage selection, which contains feature-searching methods and prediction-model development based on logistic regression (LGR), random forest (RF), classification regression tree (CART), extreme gradient boosting (XGBoost), and multivariate adaptive regression splines (MARS). The model receives as input several preoperative medical factors and their characteristics. To find the correct factors that affect the outcomes and reduce distortion, model performance relies on feature selection (Nguyen, 2010).

There were three purposes of this retrospective population-based study. The first research object was to analyze older adults’ survival rate after CABG surgery within a 10-year follow-up. Second, we used different feature-selection methods to investigate which risk factors were crucial variables that could affect survival. Lastly, we aimed to determine the best prediction survival model for older adults receiving CABG procedures, and to identify the associated factors in the prediction model that determine surgery risk factors.

## 2. Materials and Methods

### 2.1. Data Source

There are around 23 million people in Taiwan. The National Health Insurance Research Database (NHIRD) enrolls nearly 99% of Taiwanese enrollees in the National Health Insurance (NHI) program [9]. NHIRD contains the personal information of patients who participate in the NHI program, including outpatient and inpatient information, and surgical procedure codes, and it enables the continuous tracking of all claimed records from each patient. The diagnosed codes were International Classification of Diseases, Ninth Revision; Clinical Modification (ICD-9-CM); the Tenth Revision (ICD-10-CM) in Taiwan was fully adopted from 1 January 2016. According to the abovementioned advantages, the NHIRD provides complete and comprehensive long-term follow-up for each patient. Demographic ID information in NHIRD was anonymized and deidentified. This study was exempted from a full ethical review by the Fu Jen Catholic University ethics institutional review board in Taiwan (C108121), and the requirement to obtain informed consent was waived.

### 2.2. Study Population

To understand the important factors that affect older patients’ survival rate after CABG surgery, this retrospective cohort study enrolled patients over 65 years old from 1 January 2008 to 31 December 2009, from the NHIRD, Taiwan. We selected patients who had first undergone CABG operation (the operation code of only one anastomosis vessel is 68023A and 68023B, 68024A and 68024B are 2 vessels, and 68025A and 68025B are 3 diseased vessels). CABG’s initial surgery date was used as the index date to ensure that this study focused on older individuals; patients under 65 years old (*n* = 3533) were excluded. We also excluded those who had had CABG surgery before the index year (between 2002 and 2007; *n* = 39), had died in the hospital (*n* = 434), and those with missing information (*n* = 5). According to these criteria, a total of 4162 patients undergoing CABG surgery were divided into two groups, dead and alive patients ≥65 years old, between 1 January 2008 and 31 December 2009 (Figure 1).

### 2.3. Comorbidities and Variable Definitions

In this research, the baseline characteristic variables were sex, Charlson comorbidity index (CCI) score, number of anastomosis vessels, and patient comorbidities (Appendix A) including: hypertension, hyperlipidemia, diabetes mellitus (DM), congestive heart failure (CHF), peripheral vascular disease (PVD), coronary artery disease (CAD), chronic obstructive pulmonary disease (COPD), myocardial infarction (M), chronic kidney disease (CKD), end-stage renal disease (ESRD), and stroke. Blood transfusion (94001C, 94002C, 4013C, 94015C, 94003C), mechanical ventilation (57001B, 57002B, 57003B) in the preoperative one year, and CHA2DS2-VASc score [10,11] were also included. CHA2DS2-VAS was calculated for each research patient using a history of hypertension, diabetes mellitus, congestive heart failure, and vascular disease. Age between 65 and 74 years old, and female gender were 1 point. Two points were assigned for a history of ischemic stroke and transient ischemic attack (ICD-9-CM codes: 433–438; ICD-10-CM: I63.0–9, G45.9) or age ≥ 75 years old.

The date of comorbidities was defined as the date before the index date, which could be traced back to 2002–2007. Primary outcomes were overall survival rate of older adults after the CABG procedure, and cause of death was provided by the NHIRD death registry data. Patients in this study were all followed up from the index date until the date of death or the end of the research (31 December 2018).

### 2.4. Feature-Selection and Machine-Learning Prediction Models

The hospital must update each patient’s information every day. After long-term accumulation, much medical information is accumulated. We also used the NHIRD to determine key factors that affect the survival of older adults from the first CABG surgery. The medical records contained numerous items. Therefore, before making predictions, features were reduced through feature selection (FS), an essential preprocessing step [12].

However, models have different abilities to predict survival. Some studies used machine methods for an early diagnosis of bipolar disorder, prostate-cancer-specific survival, erectile dysfunction, CKD, and medical cost [13,14,15,16,17]. This research used multiple-stage selection methods to uncover potential collinearity among variable subsets and evaluate the response variable’s predictive performance. After that, we used a fivefold cross-validation process to verify the model of LGR, RF, CART, XGBoost, and MARS (for classification or continuous variables) to compare the predicted performance with all variables and evaluate the classification results after feature selection per classification method [18,19]. The classification model’s performance indicators were mean accuracy, kappa, sensitivity, specificity, and area under the ROC curve (AUC). The evaluation performance of the AUC value was defined by Hosmer et al. [17]: AUC ≥ 0.9, outstanding discrimination; 0.8 ≤ AUC < 0.9, good discrimination; 0.7 ≤ AUC < 0.8, acceptable/fair discrimination; 0.6 ≤ AUC < 0.7, poor discrimination; and AUC < 0.6, no discrimination [13]. The greater the accuracy, sensitivity, specificity, and kappa values are, the better the model is.

In this research, we used five different machine-learning methods to construct predictive models and conducted the best feature selection for evaluating the mortality of the CABG patients.

#### 2.4.1. LGR

Logistic regression is a classical prediction method suitable for predicting general binary classification problems. The central concept of LGR is the natural logarithm of an odds ratio by logit [20]. It is used to analyze the relationship between dependent and independent variables. The predicted variable Y has only two possibilities: yes (1) and no (0).

#### 2.4.2. RF

Random forest (RF) is an ensemble method, and the classifier in the original RF algorithm is a classification and regression tree (CART) that is based on the bagging algorithm and bootstrap aggregation. It randomly selects variables to split when the CART tree grows [21]. The out-of-bag (OOB) error of random forest is the average error of each weak sample using an approximate test error to measure performance [22]. Lastly, each tree was based on node impurity to improve the amplitude of the random forest and find out the importance of variables.

#### 2.4.3. MARS

MARS is a nonparametric statistical method developed by physicist Friedman et al. (1991) [23]. It is flexible regression processing that can automatically create a criterion model and separate linear-regression slopes to process multiple complex data and establish prediction models.

Approximated nonlinearity is adopted using separate linear-regression slopes in different intervals of the independent variable space. For the best MARS model, the first stage uses a forward algorithm to construct many possible basic functions and corresponding knots to initially overfit the data. We used the generalized cross-validation criterion (GCV) to generate the best combination in the second stage [22].

MARS can also use dummy variables to deal with missing values, and it does not need to assume the distribution of demand functions and errors.

#### 2.4.4. CART

Breiman et al. developed the classification and regression-tree algorithm in 1984 [24]. In the process of the CART algorithm, a series of rules are generated through recursion. First, CART builds a maximal tree to divide the two subsets into left and right through binary splits, and calculates the impurity by using the Gini index under each attribute segmentation. Nodes and leaf nodes start from the root during analysis. The smallest Gini index is used to determine segmented attributes and values. Then, the parent node can divide two exclusive children from each node, and iteratively calculate until the whole decision tree stops growing and is constructed [22].

#### 2.4.5. XGBoost

The algorithm applied by XGBoost is a gradient-boosting decision tree (GBDT) that can be used for both classification and regression problems [25]. The greedy method optimizes the maximal gain of the objective function during the construction of each tree layer. The idea of the algorithm is to continuously add trees and perform feature splitting to grow a tree. Each time a tree is added, it learns a new function to fit the residual of the last prediction. 

Lastly, multiple learners are added together to make the final prediction, and the accuracy rate is higher than that of a single one. To solve overfitting, XGBoost controls the complexity of the model by using regularization terms, and objective function optimization uses the second derivative of the Taylor expansion loss function to compute pseudoresiduals [22].

### 2.5. Statistical Analysis

Both cohorts were stratified into two groups (dead and alive) and compared using Pearson’s chi-squared tests for categorical variables. Demographic data at baseline presented numbers and percentages as *n* (%). Independent sample t-tests assessed continuous variables as means and standard deviations (mean ± SD) to compare the difference. All significance thresholds were associated with 2-tailed *p* values < 0.05. Data extraction was performed using SAS version 9.4 (SAS Institute Inc., Cary, NC, USA). Variable selection and model establishment was carried out with R statistical software (R studio 3.5.1; http://www.r-project.org (accessed on 12 January 2021)).

## 3. Results

### 3.1. Demographic Characteristics of Study Population

The demographic data and comorbidities of the patients who accepted their first CABG surgery are listed in Table 1. We included ≥65 year-old adults who had fulfilled the criteria from 1 January 2008, to 31 December 2009, in the Taiwan NHIRD. The dead group was 2272 (69.98%), and the alive group was 1456 (71.09%). In comparison, male patients had higher mortality than that of female patients.

Statistically significant results were demonstrated for the dead and alive groups. The mean follow-up periods were 4.42 ± 3.14 and 10.05 ± 0.57 years (*p* < 0.001), respectively, and the other data were as follows, as described in the brackets: CHA2DS score (4.21 ± 1.67 vs. 3.30 ± 1.57, *p* < 0.001), diabetes (65.01 vs. 50.76, *p* < 0.001), myocardial infarction (52.02 vs. 38.46, *p* < 0.001), liver cirrhosis (2.2 vs. 0.69, *p* < 0.001), peripheral vascular disease (PVD; 23.81 vs. 17.03, *p* < 0.001), congestive heart failure (CHF; 60.96 vs. 38.67, *p* < 0.001), intracranial bleeding (2.33 vs. 0.96, *p* = 0.002), atrial fibrillation (AF; 15.32 vs. 10.92, *p* < 0.001), transient ischemic attack (TIA; 41.86 vs. 29.12, *p* ≤ 0.001), chronic kidney disease (CKD; 25.18 vs. 8.86, *p* ≤ 0.001), acute coronary syndrome (ACS; 65.58 vs. 55.63, *p* < 0.001), chronic obstructive pulmonary disease (COPD; 45.91 vs. 38.32, *p* < 0.001), stroke (41.68 vs. 29.05, *p* < 0.001), cancer (7.22 vs. 4.53, *p* < 0.001) and CCI scores (3.86 ± 2.40 vs. 2.59 ± 1.93, *p* < 0.001).

The surgical variables were significantly different in terms of cost (TWD 611,701 ± 488,753 vs. TWD 394,843 ± 165,389, *p* < 0.001), the average diameter of anastomosis vessels (2.64 ± 0.72 vs. 2.79 ± 0.77, *p* < 0.0001), the length of stay (25.59 ± 14.77 vs. 18.29 ± 9.15, *p* < 0.001), blood transfusion (10.89 ± 14.68 vs. 7.23 ± 5.31, *p* < 0.001), and mechanical ventilation (7.16 ± 13.90 vs. 2.76 ± 3.09, *p* < 0.001). In addition, variables of 1 year before surgery, such as the mean number of outpatient department visits (37.70 ± 23.34 vs. 32.36 ± 20.13, *p* < 0.001), emergency department visits (2.55 vs. 0.96, *p* = 0.0006), hospitalization visits (1.91 ± 1.34 vs. 1.45 ± 0.82, *p* < 0.0001), the mean bag of blood transfusion (13.34 vs. 4.60, *p* = 0.0006), the length of mechanical ventilation (11.09 vs. 3.85, *p* < 0.001), and medical cost (155,186 ± 197,087 vs. 91,439 ± 98,235, *p* < 0.001), were also statistically significantly different between the dead and alive groups of older adults who had undergone first CABG surgery.

### 3.2. Results of Feature Selection on CABG

To determine which risk factors could predict survival among older CABG patients, we used different feature-selection methods to determine them. Ranking first was the most important. A total of 72 variables were included in this study, and each variable had its ranking in 5 different methods after filtering (Table 2)—the studied characteristics included surgical, recent 1-year variables, and the patient’s baseline. LGR selected 17 variables. RF selected a total of 11 variables. CART chose nine variables. XGBoost and MARS both selected seven variables. Among those methods, LOS, CHA2DS2 score, and CKD were only selected by CART. CART, XGBoost, and MARS all selected the risk factors of surgical cost, patient’s age, renal disease, and CCI score as essential variables.

Through different variable-selection algorithm methods, we could make predictions with these variable combinations.

### 3.3. Performance of Different Prediction Models

Lastly, we used the results of different feature-selection methods and nonfeature selection to produce five different prediction models: LGR, RF, CART, MARS, and XGBoost. In order to predict survival, the ability of each model was an independent validation dataset. The results showed that, without variable selection (72 variables), the predictive ability of XGBoost was the best (accuracy: 0.7225) among the five models (as shown in Table 3). LGR, RF, and CART individually used 17,119 variables. XGBoost had the best predictive ability (accuracy: 0.7131) and only required seven variables. The best forecasting ability among these five methods was logistic regression (accuracy: 0.7184). We also added three risk factors to the variable selections of XGBoost and MARS—CHA2DS score, acute pancreatitis, and AKF—for further predictive analysis. Adding these three variables can improve the ability of prediction models. Overall, the feature-selection method opted for XGBoost, with surgical cost, CCI scores, age, renal disease, diabetes, CHF, ulcer disease, and three risk factors (AKF, acute pancreatitis, and CHA2DS2-VAS score). The average accuracy for MARS was 0.7225; MARS was ranked as the best and only needed ten variables.

## 4. Discussion

This population-based cohort study was based on NHIRD, which is the largest observational database from Taiwan. The strengths of using NHIRD are as follows: (1) it included various individual medical information; (2) each patient could be tracked for a long-term follow-up; (3) it could show current diagnostic and therapeutic modes in the real world. The purpose of the research was to find the risk factors that could predict survival rates with different combinations of feature-selection methods and prediction models. We evaluated the survival to discharge and risks factors of older adults after the first CABG from 2008 to 2009 and followed up to 10 years. Our study showed that, without variable selection, XGBoost had the best predictive ability. By selecting XGBoost and adding the CHA2DS score, acute pancreatitis, and acute kidney failure for further predictive analysis, MARS had the best prediction performance and only needed 10 variables.

Previously, most studies focused on chronic or vascular diseases that had been acquired before the CABG surgery [26]. No known study investigated using preoperative and perioperative variables as predictor factors for long-term survival probability. A previous history of DM and CKD is a decisive risk factor for cardiovascular diseases, such as CAD and CHF. In part, most are contributed from aging [5,27,28], MI, AF, chronic renal failure, abnormal renal function, and renal failure have higher mortality after CABG [6,26,29,30,31]. Liu et al. found that ≥65 age, the female sex, diabetes, congenital heart disease, hypertension on Levels 2 and 3, and using private insurance contributed to a higher risk of readmission [1]. The score of CHA2DS2-VASc was employed as a risk-measurement tool; it was recorded in treatment guidelines for stroke prevention and is a factor for predicting stroke. Tian et al. suggest that CHA2DS2-VASc score should be on the clinical application [10]. This study demonstrated two significant findings: first, preoperative 1-year and perioperative variables are significant predictors. Second, after applying machine-learning variable screening and prediction methods, it is clearer to identify which variables could affect survival. Furthermore, we could also use fewer factors to achieve good predictive ability. Our study’s limitations are the lack of clinical lab data, such as family history, and detailed health-check values.

## 5. Conclusions

On the basis of our research, we developed multiple-stage frameworks to build a survival model for predicting the mortality of older adults who had undergone their first CABG. The advantages of this study are that it is innovative and practical in clinical research. Furthermore, we could achieve better prediction with only 10 variables. This could help clinicians make decisions more quickly and encourage patients towards earlier healthcare management.

## Figures and Tables

**Figure 1 healthcare-09-00547-f001:**
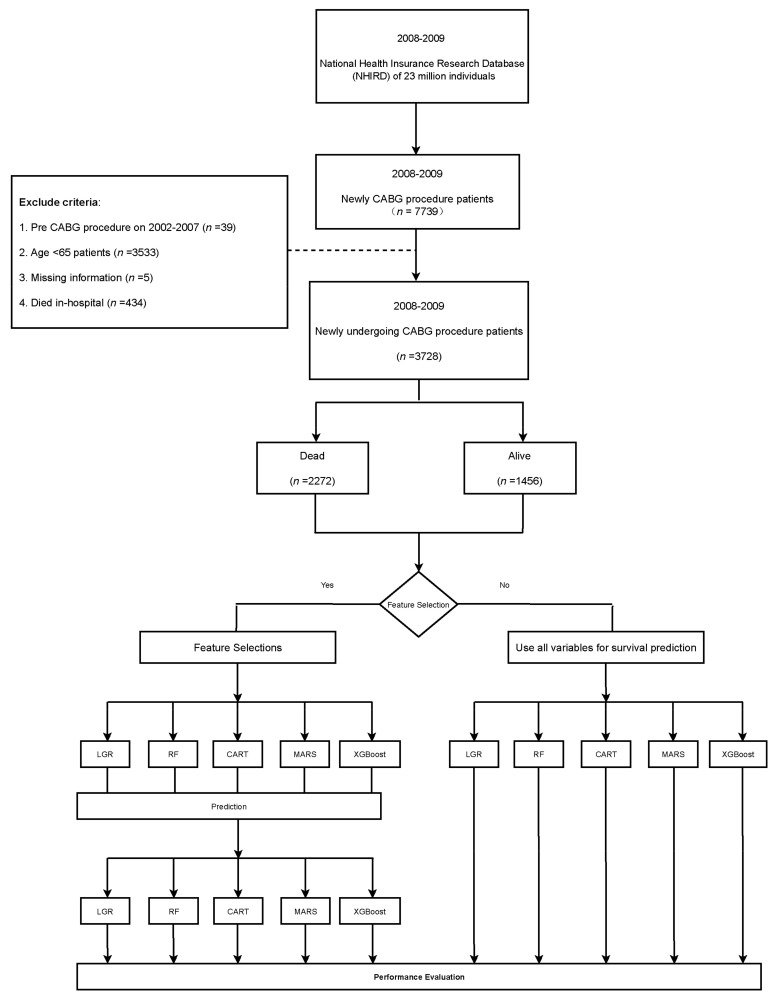
Patient selection and further analysis of 3728 older adult patients who had undergone first-time coronary artery bypass surgery grafting (CABG) between 2008 and 2009.

**Table 1 healthcare-09-00547-t001:** Demographic features of older CABG adults in Taiwan from 2008 to 2009.

Variables		≥65 Dead (*n* = 2272)		≥65 Alive (*n* = 1456)		*p*-Value
		*n*	%	*n*	%	
**Sex**	Female	682	30.02	421	28.91	0.471
	Male	1590	69.98	1035	71.09	
Age, mean (SD), y		74.30 (5.60)		71.27 (4.78)		<0.001
Follow up years, Mean (SD)		4.42(3.14)		10.05 (0.57)		<0.001
Follow up years, Median		4.22		10.02		-
CHA2DS score, mean (SD)		4.21 (1.67)		3.30 (1.57)		<0.001
**Comorbidities**						
DM		1477	65.01	739	50.76	<0.0001
Hypertension		624	27.46	379	26.03	0.335
Hyperlipidemia		1522	66.99	1056	72.53	<0.001
MI		1182	52.02	560	38.46	<0.001
Liver cirrhosis		50	2.2	10	0.69	<0.001
CHF		1385	60.96	563	38.67	<0.001
CAD		2222	97.8	1435	98.56	0.098
PVD		541	23.81	248	17.03	<0.0001
Acute pancreatitis		43	1.89	21	1.44	0.301
Malignant dysrhythmia		104	4.58	58	3.98	0.385
Intracranial bleeding		53	2.33	14	0.96	0.002
AF		348	15.32	159	10.92	<0.001
TIA		951	41.86	424	29.12	<0.0001
CKD		572	25.18	129	8.86	<0.0001
ACS		1490	65.58	810	55.63	<0.0001
COPD		1043	45.91	558	38.32	<0.0001
Stroke		947	41.68	423	29.05	<0.0001
Cancer		164	7.22	66	4.53	<0.001
**CCIS scores**	0	75	3.3	139	9.55	<0.0001
	1	269	11.84	330	22.66	
	2	383	16.86	362	24.86	
	3	424	18.66	239	16.41	
	4	341	15.01	165	11.33	
	5	275	12.1	115	7.9	
	6+	505	22.23	106	7.28	
Mean (SD)		3.86 (2.40)		2.59 (1.93)		<0.0001
**Surgical Variables**						
Anastomosis vessels, mean (SD)		2.64 (0.72)		2.79 (0.77)		<0.001
Length of stay (LOS), mean (SD)		25.59 (14.77)		18.29 (9.15)		<0.001
Blood transfusion, (Bag), mean (SD)		10.89 (14.68)		7.23 (5.31)		<0.001
Mechanical ventilation, (Day), mean (SD)		7.16 (13.90)		2.76 (3.09)		<0.001
Surgical cost		611,701 (488,753)		394,843 (165,389)		<0.001
**One Year Before Surgery**						
Outpatient visits, mean (SD)		37.70 (23.34)		32.36 (20.13)		<0.001
Hospitalization, mean (SD)		1.91 (1.34)		1.45 (0.82)		<0.001
ED visits, mean (SD)		58	2.55	14	0.96	<0.001
Blood transfusion, (Bag), mean (SD)		3.83 (3.69)		4.09 (4.87)		0.636
Mechanical ventilation, (Day), mean (SD)		5.55 (13.48)		3.93 (4.05)		0.373
Medical cost (related cardiology department), mean (SD) (thousand NT$)		81,957 (107,098)		60,969 (80,674)		<0.0001
Medical cost (thousand NT$)		155,186 (197087)		91,439 (98,235)		<0.0001

CCIS = Charlson comorbidity index score; SD: standard deviation; ED: Emergency departmen; MI: Myocardial infarct; CHF: Congestive heart failure; CAD: Coronary artery disease; PVD: Peripheral vascular disease; AF: Atrial fibrillation; TIA: Transient ischemic attack; CKD: Chronic kidney disease; ACS: Acute coronary syndrome; COPD: Chronic obstructive pulmonary disease ; AKF: Acute kidney failure ; DM: Diabetes mellitus.

**Table 2 healthcare-09-00547-t002:** Ranking of essential variables of older CABG adults.

Variables	LGR (17 Variables)	RF (11 Variables)	CART (9 Variables)	MARS (7 Variables)	XGBoost (7 Variables)
**Surgical Variables**					
Blood transfusion, (Bag), mean	1				
Length of stay (LOS), mean			4		
Surgical cost			3	1	1
**One Year Before Surgery**					
ED visits, mean	4	6			
Outpatient visits, mean	15				
Hospitalization, mean		3			
Mechanical ventilation, (Day), mean	16	7			7
Blood transfusion, (Bag), mean		1			
Medical cost			8		6
**Baseline**					
Age		11	5	3	2
CHF	7	4		6	5
CKD			7		
ACS	12				
CAD	2				
CCI score			9	2	3
COPD	11				
PVD	14				
Diabetes mellitus		5		5	
Renal disease			1	4	4
Major illness	8				
Ischemic stroke	3				
CHA2DS2 scores			2		
Ulcer disease	17			7	
Hypertension	6				
Hyperlipidemia		2			
AKF	13				
Acute pancreatitis	10				
Connective tissue disease	9	8			
Moderate or severe renal disease	5	9	6		
Moderate or severe liver disease		10			

**Table 3 healthcare-09-00547-t003:** Performance evaluation of prediction models on nonselection and after feature selection.

	Method	Accuracy	Kappa	Sensitivity	Specificity	AUC
Overall (72 variables)	LGR	0.7198	0.4427	0.6711	0.7939	0.7926
	RF	0.7077	0.3965	0.7355	0.6655	0.7784
	MARS	0.7104	0.4294	0.6444	0.8108	0.7890
	CART	0.6930	0.3360	0.8111	0.5135	0.7031
	XGBoost	**0.7225**	0.4394	0.7044	0.7500	0.7934
LGR selection (17 variables)	LGR	0.6179	0.2752	0.4888	0.8141	0.6981
	RF	**0.6260**	0.2829	0.5177	0.7905	0.6912
	MARS	0.6219	0.2771	0.5088	0.7939	0.6917
	CART	0.5911	0.2292	0.4533	0.8006	0.6576
	XGBoost	0.6246	0.2845	0.5044	0.8074	0.6977
RF selection (11 variables)	LGR	0.6876	0.3960	0.5866	0.8412	0.7784
	RF	0.6916	0.3937	0.6244	0.7939	0.7637
	MARS	0.6890	0.3817	0.6444	0.7567	0.7675
	CART	0.6930	0.3360	0.8111	0.5135	0.7031
	XGBoost	**0.6983**	0.4161	0.5977	0.8513	0.7790
CART selection (9 variables)	LGR	0.7091	0.4009	0.7311	0.6756	0.7624
	RF	0.6554	0.3464	0.5200	0.8614	0.7557
	MARS	0.7091	0.3954	0.7488	0.6486	0.7653
	CART	0.6930	0.3360	0.8111	0.5135	0.7031
	XGBoost	**0.7131**	0.4062	0.7444	0.6655	0.7652
MARS selection (7 variables)	LGR	0.6876	0.3960	0.5866	0.8412	0.7784
	RF	0.6916	0.3937	0.6244	0.7939	0.7637
	MARS	0.6890	0.3817	0.6444	0.7567	0.7675
	CART	0.6930	0.3360	0.8111	0.5135	0.7031
	XGBoost	**0.6983**	0.4161	0.5977	0.8513	0.7790
XGBoost selection (7 variables)	LGR	**0.7184**	0.4186	0.7444	0.6790	0.7739
	RF	0.6903	0.3800	0.6600	0.7364	0.7453
	MARS	0.7131	0.4096	0.7333	0.6824	0.7683
	CART	0.6930	0.3360	0.8111	0.5135	0.7031
	XGBoost	0.7104	0.4212	0.6733	0.7668	0.7763
XGBoost selection and 3 risk factors (10 variables)	LGR	0.6890	0.3937	0.6044	0.8175	0.7807
	RF	0.7037	0.4008	0.6911	0.7229	0.7727
	MARS	**0.7225**	0.4233	0.7600	0.6665	0.7831
	CART	0.6930	0.3360	0.8111	0.5135	0.7031
	XGBoost	0.6970	0.4069	0.6200	0.8141	0.7845
MARS selection and 3 risk factors (10 variables)	LGR	0.6916	0.3964	0.6155	0.8074	0.7780
	RF	0.6836	0.3806	0.6088	0.7972	0.7629
	MARS	0.7024	0.3998	0.6844	0.7297	0.7722
	CART	0.6930	0.3360	0.8111	0.5135	0.7031
	XGBoost	**0.7077**	0.4190	0.6600	0.7804	0.7806

Abbreviations: LGR: logistic regression; RF: random forest; CART: classification and regression tree; MARS: multivariate adaptive regression splines; AUC: area under the curve; XGBoost: extreme gradient boosting.

## Data Availability

Data presented in this study are not available on request from the corresponding author. Due to the General Data Protection Regulation, the data presented in this research are not publicly available.

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
