# Peer review of "Machine-Learning Techniques for Feature Selection and Prediction of Mortality in Elderly CABG Patients"

_healthcare, 2021, doi:10.3390/healthcare9050547_

Round 1

Reviewer 1 Report

The submitted draft shows the big data analysis for Taiwan for detection of Mortality in elder CABG Patients. The presented results may benefits the readers with the information about the trends. For publication it would benefits the readers if more details about prediction models are included for LGR, RF, CART, MARS, XGboost. The criterion for the selection of variables needs to be detailed further. 

Author Response

We would like to thank the editor and the anonymous reviewers for your very helpful and valuable comments on our manuscript entitled “Machine Learning Techniques for Feature Selection and Prediction of Mortality in elder CABG Patients”. We have revised the manuscript according to the reviewers’ comments that made this article more readable. The main changes in response to the reviewing results are listed below.

Reviewer 2 Report

This paper presented a study on mortality prediction in elder CABG patients. To identify critical risk factors for CABG patients is helpful for clinical decisions and general healthcare. This study used a large database and provided interesting analysis.  Comments: 1. It is not clear how long every patient was tracked after the CABG procedure. In other words, is this study interested in predicting mortality of patients in 3 years, 5 years or 10 years? Information about this is very important. When the time gets longer, the features about patients are more outdated, and patients are more likely to die due to issues other than CABG- related issues. 2. Is the mortality defined as all-cause mortality or mortality just caused by cardiovascular issues? 3. It is not clear how feature selection is conducted using different algorithms. These algorithms can select features based on customized criterion. Information is missing about the criterions used in this study. 4. The study provided 5 different sets of important risk factors selected by 5 algorithms respectively. The 5 sets have low concordance with each other. In a clinical scenario, how do we understand the results regarding the risk factors for elder CABG adults? For example, blood transfusion was deemed as the first important factor by LGR, but was deemed not important at all by the other 4 algorithms. So is it important or not?

Author Response

(The authors gave the same response as above.)
